# Comparing Abstraction in Humans and Large Language Models Using Multimodal Serial Reproduction

## Abstract

Humans extract useful abstractions of the world from noisy sensory data. Serial reproduction allows us to study how people construe the world through a paradigm similar to the game of telephone, where one person observes a stimulus and reproduces it for the next to form a chain of reproductions. Past serial reproduction experiments typically employ a single sensory modality, but humans often communicate abstractions of the world to each other through language. To investigate the effect language on the formation of abstractions, we implement a novel multimodal serial reproduction framework by asking people who receive a visual stimulus to reproduce it in a linguistic format, and vice versa. We ran unimodal and multimodal chains with both humans and GPT-4 and find that adding language as a modality has a larger effect on human reproductions than GPT-4's. This suggests human visual and linguistic representations are more dissociable than those of GPT-4.

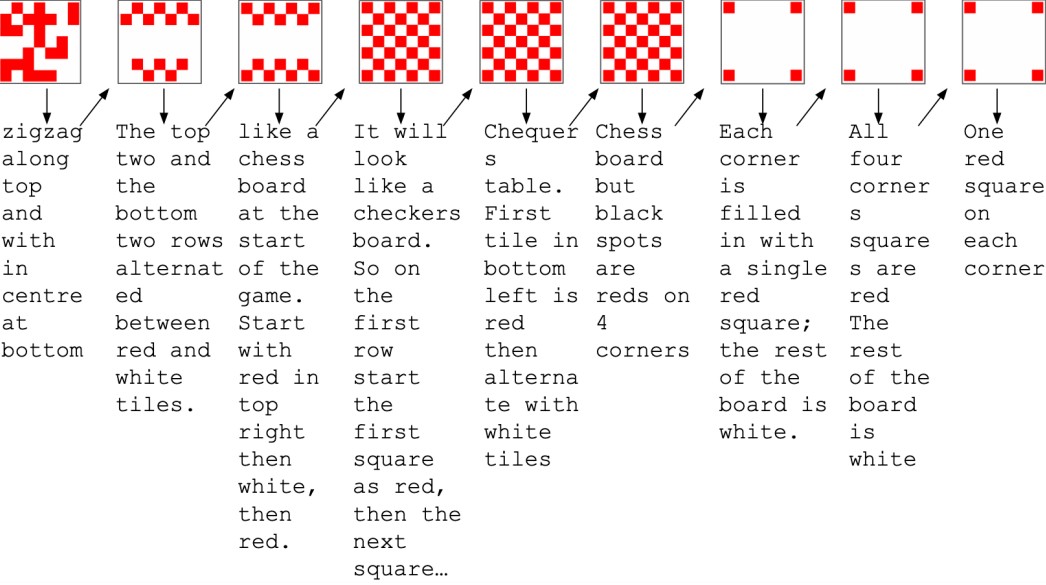

Figure 1: **Example Multimodal Serial Reproduction Chain in Humans.** One participant sees a stimulus and transmits a language description of the stimulus. The next participant sees a language description and produces a stimulus matching the description. The chain alternates between vision and language.

# 1 INTRODUCTION

Abstraction is a hallmark of human intelligence that helps us make sense of our complex environment. The ability to form abstractions has been proposed as a key component of human cognition, and necessary for artificial intelligence to exhibit the same ability to generalize from limited data (Lake et al., 2017). Large Language Models (LLMs) are sophisticated, high-performing artificial intelligence systems that have emergent properties some claim may rival human-level general intelligence (Wei et al., 2022; Bubeck et al., 2023). However, others have pointed out inconsistencies in their reasoning abilities (Mitchell et al., 2023). The difficulty of assessing these capacities highlights the need to develop rigorous experimental tools for probing abstraction in humans and machines (Lake et al., 2017; Mitchell et al., 2023; Kumar et al., 2023).

Abstraction involves capturing the essential details of incoming information that will help us generalize to future experiences while discarding less useful information (Giunchiglia & Walsh, 1992). The choice of what to focus on and what to ignore can be a reflection of our prior beliefs or expectations. Serial reproduction (Bartlett, 1932) is a method used to elicit such priors in human perception and memory through a telephone-game-like experiment (Xu & Griffiths, 2008). People are asked to pass a piece of information to one another in sequence, with each person reproducing the information from memory for the next person. Studying how the information changes as it is passed along the chain of people can be used as a window into their prior beliefs.

When using serial reproduction to study abstraction, one must consider that abstractions are not only a noisy compression of a stimulus, but they are also formed to communicate information to others (Tessler & Goodman, 2019; Tessler et al., 2021). In humans this is often done through language (Lupyan & Bergen, 2016). The extent to which our abstractions are influenced by language is a central but unanswered question in cognitive science (Quilty-Dunn et al., 2023; Kumar et al., 2022; Lupyan et al., 2007). Most experiments involving human participants constructing a serial reproduction chain (e.g. (Langlois et al., 2021; Anglada-Tort et al., 2023)) employ only one sensory modality. Therefore, incorporating language as a transmission modality in *multimodal* serial reproduction potentially provides a way to understand its influence on human abstractions.

In this work, we explore how adding language to a visual serial reproduction chain influences the output of that chain, comparing human participants and GPT-4 (a contemporary LLM with visual capabilities; Achiam et al. 2023). To simulate multimodal serial reproduction, participants who observed a visual stimulus were asked to produce a textual stimulus for the next participant and vice versa. As a result, the multimodal serial reproduction chain alternates between the visual and language modalities. We present a theoretical analysis showing that comparing unimodal and multimodal chains can allow us to assess whether distinct priors are being used to make inferences in different modalities.

To establish this comparison, we collected data from both human participants and GPT-4 in two serial reproduction paradigms, one unimodal and one multimodal. This allows us to compare emergent distributions from vision-only vs. hybrid vision-language chains to see the impact of transmission through multiple modalities. Our results show that transmission through both language and vision has a significant impact on the level of abstraction demonstrated within human participants' chains but does not have as significant impact on GPT-4, suggesting that GPT-4, unlike humans, naturally relies on language representations by default even in a vision-only paradigm.

# 2 METHODS

## 2.1 THEORETICAL FRAMEWORK

Bartlett 1932 proposed serial reproduction to study how bias from people's previous experiences influences how they perceive new experiences. In a unimodal serial reproduction study, the original stimulus is presented to the first participant, who then reproduces it for the second participant, and so on. Formally, serial reproduction can be interpreted as a Markov chain over a pair of variables $(x, \mu)$ where $x$ represents the distribution of stimuli in the world and $\mu$ represents the abstractions people infer from those stimuli, $\cdots x_t \to \mu_t \to x_{t+1} \to \mu_{t+1} \to \cdots$. A mathematical interpretation of this process used by Xu & Griffiths 2010 treats $x_t$ as a noisy stimulus, and assumes humans share a prior distribution $p_S(\mu)$ about the world. To reconstruct the observed stimulus $x_t$, humans try to estimate

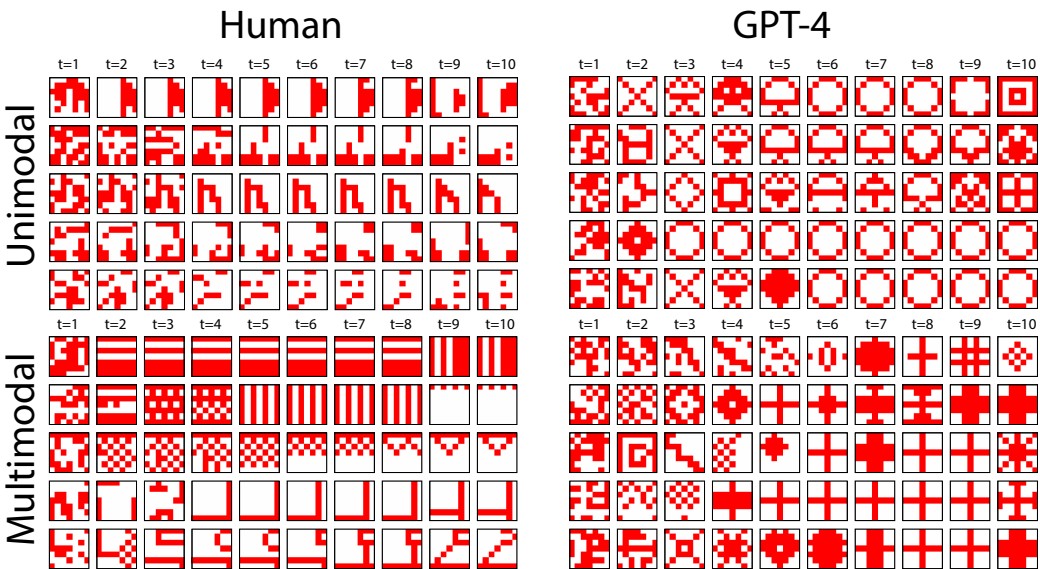

Figure 2: **Serial Reproduction Chains Across Modalities.** Five example human and GPT-4 chains for each paradigm.

the true state of the world $\mu_t$ by sampling the posterior distribution $p(\mu_t|x_t)$, and the next stimulus is then sampled from the likelihood $p_S(x_{t+1}|\mu_t)$. Under this assumption and ergodicity (i.e., that there is a finite probability for reaching any one state from another), the stationary distribution of the Markov chain over $\mu$ converges to the prior $p_S(\mu)$. Related analyses for this paradigm offer similar conclusions (Jacoby & McDermott, 2017).

Consider next a serial reproduction process in which information is transmitted bimodally, e.g., through images $x$ and language $\ell$. The process unfolds as follows: at a given iteration $t$, a human participant observes a stimulus $x_t$ and forms an abstraction $\mu_t$ about its content which they then use to form a text description $\ell_{t+1/2}$. The text description is then read by another participant who in turn forms an abstraction $\mu_{t+1/2}$ and uses that abstraction to produce a new stimulus $x_{t+1}$. In other words, we are concerned with the Markov process $\cdots \to x_t \to \mu_t \to \ell_{t+1/2} \to \mu_{t+1/2} \to x_{t+1} \to \cdots$ and want to characterize the stationary distribution of $p(x_{t+1}|x_t)$. Following Xu & Griffiths 2010, we assume the agents are Bayesian and that they use some prior knowledge to form abstractions from language and stimuli, namely $p(\mu|x) \propto p_S(x|\mu)p_S(\mu)$ and $p(\mu|\ell) \propto p_L(\ell|\mu)p_L(\mu)$ where $S$ and $L$ indicate stimulus and language, respectively. The stimulus and language priors over abstractions need not be the same (the likelihoods are by definition different because they are defined on different input). If, however, they were aligned, i.e. $p_S(\mu) = p_L(\mu)$ for all $\mu$, then propagation through the additional linguistic modality does not alter the stationary distribution over stimuli $x$. This can be verified by checking that the prior predictive distribution $\hat{p}(x) = \int p_S(x|\mu)p_S(\mu)d\mu$ satisfies the stationarity condition $\int p(x_{t+1}|x_t)\hat{p}(x_t)dx_t = \hat{p}(x_{t+1})$, similar to the unimodal case of Xu & Griffiths 2010. Discrepancies in the stationary distribution over stimuli $x$ between the unimodal and multimodal serial reproduction chains in this theoretical setup would reflect a difference in priors $p_L(\mu)$ and $p_S(\mu)$. This suggests that comparison of the stationary distributions produced by unimodal and multimodal chains is an effective way of discovering differences in the way that agents construe the world across different modalities.

## 2.2 HUMAN EXPERIMENTS

We recruited $N = 348$ participants from Prolific. Participants were required to be native English speakers to ensure high textual data quality, and they provided informed consent prior to participation in accordance with an approved institional review board (IRB) protocol.

We collected 100 chains of 10 visual steps of both unimodal and multimodal serial reproduction. Both conditions were intitialzed with the same set of randomly sampled boards. Although 10 it-

erations can be considered on the short side, sampling-based chains with people tend to converge much faster than their theoretical counterparts (Sanborn & Griffiths, 2007; Harrison et al., 2020). To compensate for the chain length, we run many different chains (Harrison et al., 2020).

In this work, we use a simple stimulus space of binary $7 \times 7$ grid patterns (Fig. 1), which has previously been used to study abstraction in humans and machines (Kumar et al., 2021; 2022; 2023) due to having a nice balance between being rich enough to elicit interesting abstractions but small enough to enable rigorous experimentation.

**Unimodal Serial Reproduction:** To implement unimodal serial reproduction in humans, at each step, the participant is asked to memorize a stimulus board for 5 seconds and tasked with reproducing the board afterwards. The new board serves as the stimulus for the next iteration of the chain. After running all the unimodal chains, we then collected language descriptions of all boards produced *post-hoc* by having a separate set of participants give board descriptions. Each participant completed up to 10 trials and was allowed to visit each chain only once (to reduce trial dependence within chains).

**Multimodal Serial Reproduction:** In multimodal serial reproduction, a participant can be shown either a stimulus board, or a string of textual description. If shown a board, then they are asked to provide an accurate textual description of the board such that the board can be reconstructed from it. If shown a string of textual descriptions, then they are asked to reproduce a board that most accurately illustrates the textual description. The new board or text will serve as the stimulus for the next iteration in the chain. Here too participants completed up to 10 trials and were allowed to participate in a given chain once.

## 2.3 MACHINE EXPERIMENTS

To study machine priors, we use GPT-4 vision (Achiam et al., 2023), a Large Language Model (LLM) with multimodal capabilities. We implement the serial reproduction chains with GPT-4 to be as close to the human experiments as possible. Just like the human experiments, we ran 100 unimodal and multimodal chains of 10 iterations.

**Unimodal Serial Reproduction:** To implement unimodal serial reproduction in GPT-4, we present it an image of a $7 \times 7$ binary grid and ask it to produce the grid in matrix form, with $1$ corresponding to red tiles and $0$ corresponding to white tiles. We then use the matrix GPT-4 produced for the next iteration's input.

**Multimodal Serial Reproduction:** In multimodal serial reproduction, GPT-4 can be shown either a stimulus board, or a string of textual description. If shown a board, then it is asked to provide an accurate textual description of the board. If shown a textual description, then it is asked to reproduce a board that most accurately illustrates the textual description. The new board or text will serve as the stimulus for the next iteration in the chain. We used the same prompt given to human participants.

## 2.4 MEASURES OF BOARD COMPLEXITY

The process of human abstraction aims to compress complex stimuli or inputs to simpler representations that enable generalization (Giunchiglia & Walsh, 1992; Kumar et al., 2022). Therefore, measuring the compressibility of the stimuli that emerge from the serial reproduction chains can be informative of the underlying abstract priors that generate them. There are many measures of board complexity, so we utilize three measures from the work of Nath et al. 2023, which are specifically tailored to binary grid stimuli:

1. **Kolmogorov Complexity (KC):** a measure formalized through algorithmic information theory, defined as the length of the shortest computer program that can produce the desired stimulus. The exact computation is intractable, so most empirical methods estimate an upper bound. Nath et al. 2023 use the Block Decomposition Method (Zenil et al., 2018), which breaks the grid stimulus into $4 \times 4$ blocks and uses theoretically defined complexity measures of each binary $4 \times 4$ block.

2. **Shannon Entropy:** a measure of the information content/complexity (Shannon, 1948) of the grids using its distribution of red and white tiles: $-(P(red) \log_2 P(red) + P(white) \log_2 P(white))$

3. **Local Spatial Complexity (LSC)**: the mean information gain of tiles having the same color or different colors in their adjacent tiles. This takes into account the local probabilistic spatial distribution of tiles. It is defined as $-\frac{1}{8}\sum_{d=1}^{8}\sum_{s_1=0}^{1}\sum_{s_2=0}^{1}P(s_1, s_2)_d \log_2 P(s_1|s_2)$ where $s_1$ and $s_2$ are adjacent tiles whose spatial relation is defined through $d$. There are eight possible values for $d$ corresponding to the four cardinal directions as well as four diagonals.

Each of these measures provide a slightly different window into the complexity of a board. For example, Kolmogorov Complexity is the only measure taking into account the algorithimic complexity of the board (e.g. algorithms to generate the patterns). Shannon Entropy formally measures the information-theoretic content based on the distribution of tile colors, but does not take into account spatial information. Local Spatial Complexity is an information theory measure more sensitive to spatial information because it looks at the local distribution of a tile's nearest neighbors.

# 3 RESULTS

## 3.1 QUALITATIVE BOARD DISTRIBUTION

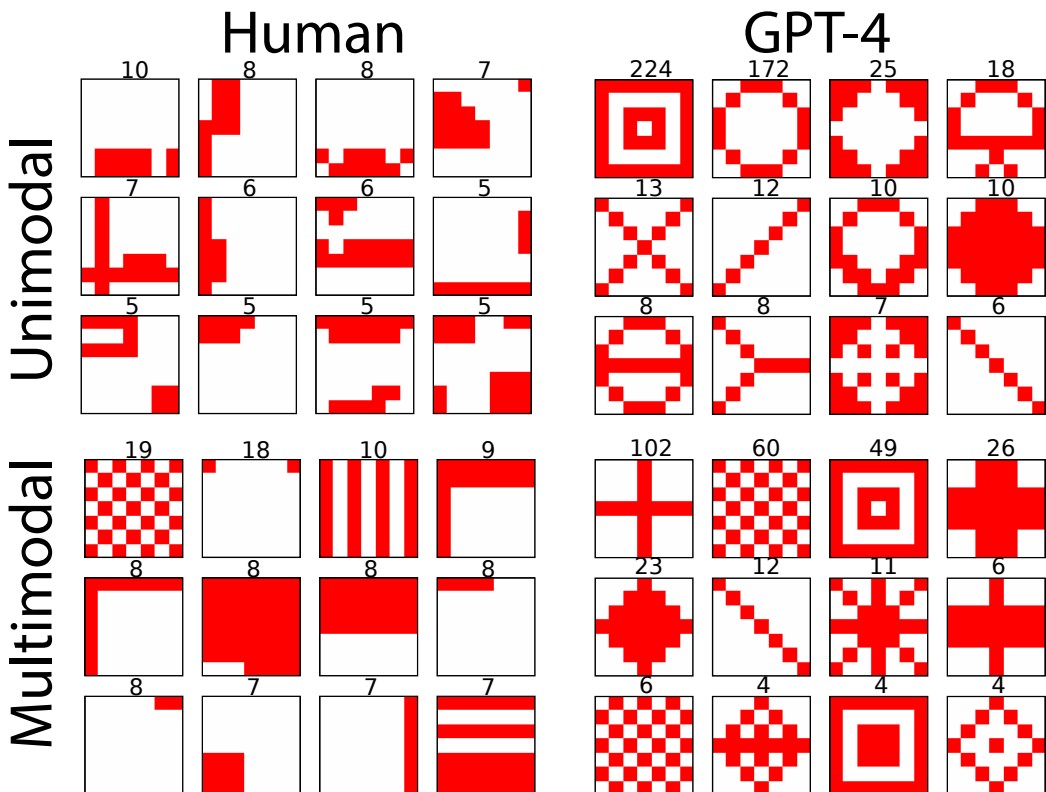

Figure 3: **Most Frequent Boards Across Conditions.** Numbers indicate the frequency of the board below it.

Fig. 3 shows the most frequent boards across the four conditions. Because there is more variance across different human participants' responses than GPT-4 responses, the largest frequencies of GPT-4 are typically higher than humans. The most frequent human multimodal boards seem to be patterns that are most easily identifiable by language, e.g., checkerboard, square shapes, and stripes (the checkerboard pattern, in particular, seems to only show up in multimodal chains for both humans and GPT-4) whereas unimodal boards are patterns that are harder to describe in language. However, in the case of GPT-4, both unimodal and multimodal patterns are more easily describable through language.

## 3.2 CHAIN DYNAMICS

Fig. 4 shows the mean distance traveled between consecutive timesteps in each of the four types of serial reproduction chains. This can be thought of as a measure of *instantaneous velocity* since it measures distance traveled within consecutive timesteps ($dt = 1$).

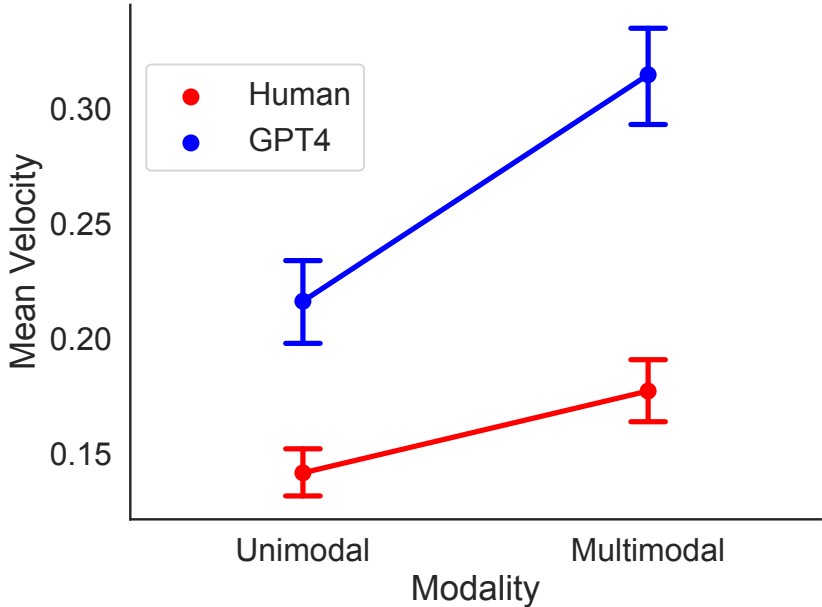

Figure 4: **Mean Chain Velocity** We computed mean instantaneous velocity of each chain by computing the hamming distance traveled between boards of consecutive timesteps. Error bars denote 95% confidence intervals across chains.

We see that multimodal chains have significantly higher velocity than unimodal chains in both conditions (human: $t(198) = 4.05, p < 0.0001$; GPT-4: $t(198) = -7.01, p < 0.0001$). Since the chief difference between these conditions is the addition of language as a bottleneck between state transitions in the chain, this shows that transmitting through language further *accelerates* the sampling of the abstraction space more. GPT-4 has significantly higher chain velocity than humans in both conditions (unimodal: $t(198) = 6.97, p < 0.0001$; GPT-4: $t(198) = 10.87, p < 0.0001$), which may suggest that GPT-4, by default, relies more on language representations than humans do.

## 3.3 BOARD COMPLEXITY ANALYSES

We looked at the distribution of complexities across human and machine chains for different conditions (Fig. 5A). Qualitatively, running a multimodal serial reproduction chain seems to have a bigger effect on mean board complexity for humans than for GPT-4 (signified by the red line having a larger slope than the blue line). To statistically evaluate this, for each complexity measure, we ran a two-way ANOVA with subject (human or GPT-4) and modality (unimodal and multimodal) as factors that account for the mean complexity of the boards (Table 1).

We find all two-way interaction effects between the subject (human and GPT-4) vs. modality (unimodal vs. multimodal) to be consistent in direction and statistically significant. As seen in Fig. 5A, this effect is driven by a tendency for there to be a greater difference between human unimodal and multimodal board complexity than GPT-4 unimodal vs. multimodal board complexity. We also found significant one-way effects of subject. The direction of this effect was consistent in all measures — GPT-4 tends to have a higher mean board complexity than humans. In addition, we also find significant one-way effects of modality (unimodal vs. multimodal), suggesting that additionally transmitting through language decreases board complexity.

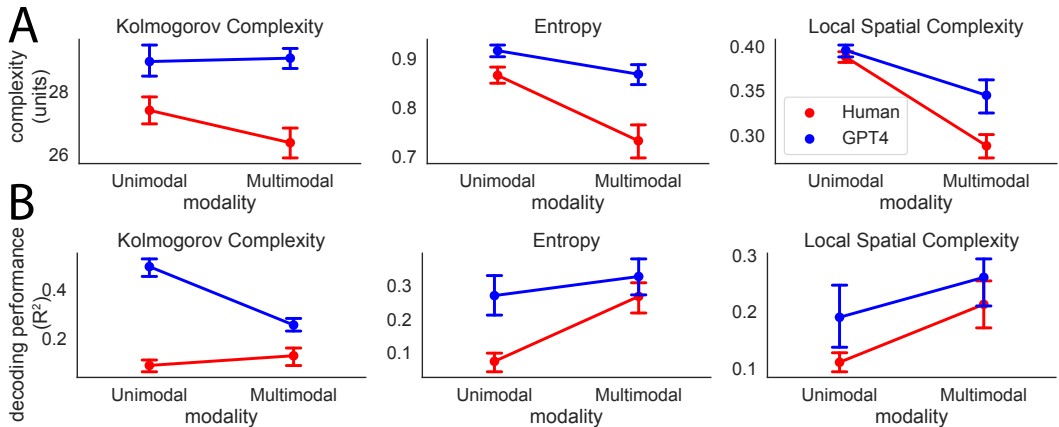

Figure 5: **Transmitting through language has a larger effect on humans than GPT-4** (A). 95% confidence intervals for complexity measures across humans and GPT-4 for both types of chains. GPT-4 boards typically have higher complexity. Multimodal serial reproduction typically reduces complexity, and this reduction is more pronounced in humans than GPT-4. (B). Decoding ($R^2$) performance for predicting board complexity from the corresponding language description's sentence embeddings. Higher performance suggests that the complexity of the boards can be represented in language. Decoding performance increases from unimodal to multimodal chains and GPT-4 boards have higher decoding performance than human boards.

Table 1: Two-way ANOVAs for Board Complexity

| Measure | Effect | $F$ | $p$ |
|---|---|---|---|
| KC | Modality | 6.99 | 0.009 |
| KC | Subject | 65.39 | <0.001 |
| KC | Interaction | 4.08 | 0.044 |
| Entropy | Modality | 65.94 | <0.001 |
| Entropy | Subject | 43.68 | <0.001 |
| Entropy | Interaction | 8.09 | 0.005 |
| LSC | Modality | 151.49 | <0.001 |
| LSC | Subject | 24.09 | <0.001 |
| LSC | Interaction | 14.63 | <0.001 |

Since the chief difference between the modality conditions (unimodal vs. multimodal) is explicit transmission through a language description, the fact that these conditions have a significantly larger effect on humans than GPT-4 may suggest that GPT-4 relies more on language-compatible representations than humans, leading to a lesser effect when explicitly forcing it to transmit through a language bottleneck. Equivalently, this may suggest that humans' abstract vision and language priors have less overlap than GPT-4's vision and language priors.

## 3.4 DECODING BOARD COMPLEXITY FROM LANGUAGE

We now employ an analysis to see if the complexity within each of these boards measured in Fig. 5A is the kind of complexity that can be represented in language (Fig. 5B). To do this, we obtained pre-trained LLM sentence embeddings of each board's corresponding language description using the SentenceTransformers package (https://www.sbert.net/, based on Reimers & Gurevych 2019). The pretrained model we used was Microsoft's MPNet (Song et al., 2020). The Sentence-Transformers model maps text into a 768 dimensional dense semantically meaningful vector space. Reimers & Gurevych 2019 do this by using a contrastive objective on a dataset of semantically-paired text where embeddings from the same pair are pushed closer and embeddings from different pairs are pushed further apart.

Table 2: Two-way ANOVAs for Language Decoding

| Measure | Effect | $F$ | $p$ |
|---------|--------|-----|-----|
| KC | Modality | 19.28 | 0.005 |
| KC | Subject | 225.70 | <0.001 |
| KC | Interaction | 65.67 | <0.001 |
| Entropy | Modality | 11.14 | 0.004 |
| Entropy | Subject | 10.24 | 0.006 |
| Entropy | Interaction | 1.12 | 0.31 |
| LSC | Modality | 35.75 | <0.001 |
| LSC | Subject | 8.84 | 0.009 |
| LSC | Interaction | 1.45 | 0.25 |

In multimodal boards, we use the language description that was obtained by a participant (or GPT-4) who viewed the board and wrote the description (see Fig. 1). In unimodal boards, we repeated this process *post-hoc* after the chains were completed by showing a separate set of participants (or GPT-4) each unimodal board and asked them to write a language description (using the same prompt as the participants who wrote descriptions in the multimodal chain condition).

We then take these embeddings and train a Ridge Regression model to predict board complexities from its corresponding sentence embedding (Fig. 5B). We use five-fold cross validation and report mean prediction accuracy ($R^2$) on the held-out test set across all five folds. The regularization parameter is tuned using a nested five-fold cross validation procedure within the training set (so, for each outer fold, the regularization parameter is tuned before the test set is ever seen). Qualitatively, we see that there is a general upward trend (more pronounced in humans than GPT-4) in decoding performance from unimodal to multimodal boards. To quantify this effect, we repeated the two-way ANOVA analyses employed in the last section, with subject (human vs. GPT-4) and modality (unimodal vs. multimodal) as factors that influence the decoding performance of the Ridge Regression model. Results are shown in Table 2.

The one-way effect of subject (human vs. GPT-4) was significant across all three measures and was consistent in direction — GPT-4 decoding performance is higher. Additionally, the one-way effect of modality (unimodal vs. multimodal) was also significant across all three measures, showing that multimodal boards have higher decoding performance than unimodal boards. The interaction (subject + modality) effect was only significant in one measure, Kolmogorov Complexity.

These results suggest two main findings. First, decoding performance of complexity from language embeddings is generally higher in multimodal chains than in unimodal chains. This suggests that the complexity of the multimodal boards, compared to unimodal boards, is more the kind of complexity that can be accounted for by language representations. Second, GPT-4 generally has a higher decoding performance than humans. Although the complexity of GPT-4 boards are generally higher than those of humans (see previous section and Fig. 5A), this complexity is the kind of complexity that is decodable by language representations.

## 4    DISCUSSION

In this work, we explored abstractions in humans and GPT-4 using a framework involving serial reproduction within a simple yet rich stimulus space of binary grid boards (Fig. 1). Previously, serial reproduction has been used to elicit human priors (e.g., Langlois et al. 2021). Humans often share abstractions of their sensory experience with each other through language (Tessler & Goodman, 2019). However, the abstractions humans build of the world can be represented through multiple modalities (Hawkins et al., 2023), and serial reproduction and similar iterative methods typically only employ a single modality. This paper presents a novel *multimodal* serial reproduction paradigm, in which people alternate between transmitting through both vision and language. This provides a way to determine the extent to which priors are shared across modalities.

We ran both unimodal and multimodal serial reproduction chains for both humans and GPT-4 (Fig. 2). Qualitatively, for humans, we found that multimodal chain samples seem much easier

to describe in language (Fig. 3). Quantitatively, we found evidence that the addition of transmission through language leads to the emergence of more compressible (and, therefore, more abstract) stimuli in the stationary distribution (Fig. 5A). Additionally, language representations are more predictive of the complexity of human multimodal boards than that of unimodal boards (Fig. 5B), suggesting that, in humans, a purely unimodal paradigm does not tap into abstractions that can be shared with language as much as a multimodal paradigm. In contrast, GPT-4 unimodal and multimodal boards both qualitatively seem easy to describe in language (Fig. 3). Quantitatively, the change in complexity across unimodal and multimodal boards is significantly less in GPT-4 than the corresponding change in humans (Fig. 5A). Language representations are more predictive of GPT-4 board complexities than human representations (Fig. 5B).

This evidence suggests that GPT-4 abstract visual representations are much closer to linguistic representations than those of humans. This may have resulted from the training paradigm for GPT-4. Although the information on how GPT-4 was trained is not fully public, many similar vision-language models are trained on large amounts of text data and jointly match images with their language descriptions during training(Radford et al., 2021), leading to a tight coupling between vision and language representations. The fact that human unimodal and multimodal boards have a significantly greater difference than those of GPT-4 (Figs. 3 and 5) suggests that humans, in contrast, have more dissociable representations between vision and language. This may be because the human visual system was first evolutionarily refined to support embodied sensation and movement within the environment (Cisek, 2019) before communicating sensory experience to other humans through language.

One limitation of this work is that we use a fairly constrained domain of two-dimensional binary grids. It is possible results could differ on more realistic visual inputs for humans as well as GPT-4's. Our multimodal serial reproduction framework can easily be extended to more realistic-looking visual domains, potentially using drawing to transmit images from language (Mukherjee et al., 2023). Likewise, our chains were relatively short, and longer chains could be useful as a control for the initial effect of mixing. This can be easily addressed by deploying larger online experiments.

It is also possible to extend our work to run hybrid serial reproduction chains with both humans and machines. This could help us see what concepts arise from joint shared abstractions between humans and machines, which will become increasingly relevant as AI systems become further incorporated into our daily lives (Brinkmann et al., 2023).

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
