# OpenReview forum: "Comparing Abstraction in Humans and Large Language Models Using Multimodal Serial Reproduction"
_ICLR.cc/2024/Workshop/Re-Align — ICLR 2024 Workshop Re-Align Poster_

### Official Review · Reviewer_V6kB · 2024-02-25
**An investigation of Human and GPT4 abstractions**

**Rating:** 2
**Fit:** 3
**Confidence:** 3

**Workshop Review:**

This paper uses a serial reproduction experiment to investigate differences between a purely visual reproduction and a multimodal linguistic and visual reproduction. The reproduction chains are analyzed in humans and in GPT-4. The authors find differences between the unimodal and multimodal cases, and also find differences between humans and GPT-4. The analyses suggest that the GPT-4 differences between the unimodal and multimodal experiments are less than the human differences between the cases, and the authors conclude that this suggests GPT-4 relies on its “linguistic” representation while the visual and linguistic representations are separable in humans.

This experiment is interesting and will likely spark conversation at the workshop, and so I recommend acceptance. However, some aspects of the experiments and analysis require fine-tuning to truly understand what is happening and communicate this well to the readers. Additional details are below which would improve the paper.


Major:
* I remain confused about how the authors are using the serial reproduction framework (which typically has been used to study prior beliefs/expectations) to also study the influence of language. The paradigm seems interesting, but the logic behind this needs to be better fleshed out – I believe this is attempted on page 3, but it is so critical to the methods that it needs to be made clear. Perhaps part of this is because it seems difficult to understand (or think about testing) what the underlying $p_L(\mu)$ would be, but perhaps that is partially the point? Maybe by looking at this comparison, one can avoid directly creating a test for $p_L(\mu)$? Walking through the logic of this would be very valuable for the reader and for others using these types of behavioral paradigms generally.
* It further seems like a control case where humans do not need to rely on a memory bottleneck for communicating the information should be investigated, especially for the multi-modal case.
* Why does the main analysis rely on measures of complexity that seem to be measured on the full chain, rather than just the end point? Typically I think of serial reproduction strategies as being interesting once they reach some stationary distribution, but it seems that early and late trials are all being analyzed the same here?
* Related to the above, I am unsure that the entropy analysis is sufficient to get at the underlying question at hand. It seems critical to measure whether some of the information from the current state is being used to produce the next state. This is somewhat asked with the “velocity” analysis, but it is not sufficiently addressed, as a high “mean velocity” could be due to a model or human just ignoring the input state, and I’m not sure that this counts as “accelerating” the sampling of the abstraction space more. Perhaps one way to control this would be to ask both models and humans to generate things on the 7x7 grid without viewing any previous sample? That seems like another way to explore the abstraction space? (Also note: the velocity analysis needs to have more methods explanation, as right now it is difficult to follow).
* In the decoding board complexity analysis, I am concerned this is biased by the fact that the GPT chains have less variance than human participants, and also that there might not be equal variance in the unimodal and multimodal conditions. This would need to be controlled in order to say anything about the decoding analysis of the entropy. For instance, if there are only a small number of unique examples, the same example might end up in both the train and test splits? Even if the analysis is *only* done on unique examples, there should be the same number of unique examples in each case to avoid one decoding being inherently easier than another.
* The concern about the decoding analysis above is in the variance among the individual samples (ie Figure 3) but another concern is that the underlying variance in the *entropy* could bias the results, even if the individual samples were controlled (for instance, if everything in a condition has the same entropy, then it is easy to predict what it should be, and this says nothing about the “language” representations). These types of controls need to be addressed before concluding that one representation is easier to code "linguistically" than other.
* Is it fair to compare chains composed of different individuals to chains from a single model? This should at minimum be addressed in the discussion, along with suggestions for how to tackle it in the future.

Minor:
* I did not let this cloud my judgment of the paper, as it is more of an existential crisis about the state of the field, but I generally question whether the cognitive science community can learn anything interesting by studying closed models when we don’t know how it was trained, what the underlying architecture is, etc. Will any results generalize to learning something interesting about either humans or machines? Or is this specifically about testing a single instance of a model (but it likely isn’t even a single instance because GPT4 is constantly being updated behind the scenes, which means the results may never be reproducible)? Is the goal of the paper to study humans or to study GPT4?
* When the GPT examples were drawn, was a new session opened for each, or did the model retain memory/state from the previous instance?

**Reason For Not Giving Higher Score:**

Many of the analyses seemed preliminary and lacked appropriate controls.

**Reason For Not Giving Lower Score:**

The behavioral experiment is quite interesting and thought-provoking, and I think the topic will spark interesting discussion at the workshop.

**Reviewer Domain:**

cognitive science

---

### Official Review · Reviewer_S8EG · 2024-02-26
**A comparative evaluation study of the visual reasoning capabilities of GPT-4 vs humans**

**Rating:** 1
**Fit:** 2
**Confidence:** 2

**Workshop Review:**

The authors define a visual reasoning task based on "telephone" game. They define two versions of the task: the first unimodal and second multimodal (vision+language). Further, the authors define 3 metrics to quantify complexity of the board.
A few comments:
From figure 3, the finding that "there is more variance across different human participants’ responses than GPT-4 responses, the largest frequencies of GPT-4 are typically higher than humans" seems important. In the multimodal task, GPT4 is restricted to extremely regular patterns as compared to humans. This finding seems to explain all the results of the paper.
Further, GPT4 seems to make larger jumps across boards and thus is probably related to the previous finding that GPT4 can only generate highly structured and thus moves from one structured board to an entirely different one.
And even further, in figure 5 GPT-4 has a higher decoding performance, probably since the boards it generates are more structured.
The main results about the paper are thus about the difference between multimodal and unimodal generation as opposed to model vs human performance, making me unsure of the fit of this paper for the workshop. It would be more appropriate for a purely cognitive science venue.
A good contribution of this study would be open-sourcing the dataset and eval task used in this study. This dataset can then be used in open-source replications of the study where more mechanistic and controlled evals can be performed.

**Reason For Not Giving Higher Score:**

A few reasons:
This is a purely behavioral eval paper and on a closed model with only API access and thus the results will be variable and harder to replicate. Thus, we do get no insight about the features or mechanisms used by the model to generate their responses. Further, all results comparing gpt-4 with humans seem to be explained by the fact that gpt-4 generates very structured boards as opposed to more stochastic boards generated by humans.

**Reason For Not Giving Lower Score:**

A good contribution of this study would be open-sourcing the dataset and eval task used in this study. This dataset can then be used in open-source replications of the study where more mechanistic and controlled eval can be performed.

**Reviewer Domain:**

neuroscience

---

### Official Review · Reviewer_Dd98 · 2024-02-28
**Interesting paper with cryptic analysis**

**Rating:** 2
**Fit:** 3
**Confidence:** 2

**Workshop Review:**

This paper tackles an interesting question of testing a hypothesis around multimodal serial reproduction. They performed a theoretical analysis to answer how distinct priors are being used to make inferences that involve unimodal and multimodal stimulus. They conclude that while humans use both language and vision reproduction chains, such multimodal information does not have as significant impact on GPT-4.

**Reason For Not Giving Higher Score:**

- Please improve the paper presentation, highlight the main takeaways, it might need some reframing.
- Please elaborate the following claims. How does the language be the bottleneck but helps to accelerate the abstraction space sampling?
> Since the chief difference between these conditions is the addition of language as a bottleneck between state transitions in the chain, this shows that transmitting through language further accelerates the sampling of the abstraction space more.
> Since the chief difference between the modality conditions (unimodal vs. multimodal) is explicit transmission through a language description, the fact that these conditions have a significantly larger effect on humans than GPT-4 may suggest that GPT-4 relies more on language-compatible repre- sentations than humans, leading to a lesser effect when explicitly forcing it to transmit through a language bottleneck.
-	Lack of explanation towards the board complexity results
-	It is not clear what authors meant as higher decoding
> GPT-4 generally has a higher decoding performance than humans.

**Reason For Not Giving Lower Score:**

- Interesting experimental setup with human participant and GPT-4 to test the unimodal and multimodal reproduction chains
- Interesting observations. Looking at Figure 3, it seems GPT4 kindly agree to a set of predeterminate patterns

**Reviewer Domain:**

machine learning

---

### Official Review · Reviewer_6QSY · 2024-02-29
**The paper presents a new framework (multimodal serial reproduction) to study abstraction in humans and machines**

**Rating:** 3
**Fit:** 3
**Confidence:** 2

**Workshop Review:**

The paper studies the role of language in forming abstractions using a novel multi-modal serial reproduction framework. The authors study this in humans and GPT-4.  They offer a theoretical analysis that demonstrates how the comparison of unimodal and multimodal chains can enable us to determine if different prior knowledge is applied in making inferences across various modalities. They use different measures to compare the unimodal and multimodal chains, these include mean chain velocity (computed by averaging the instantaneous velocity of each chain by comparing the hamming distance traveled between boards of consecutive steps), complexity measures like Kolmogorov complexity (measures the algorithmic complexity), Shannon Entropy (measures the information content/complexity), and Local Space Complexity (measures the information-theoretic measure of spatial information). The results show that multi-modal and uni-modal chains differ in their chain velocity, showing that language accelerates the sampling of the abstraction space more. Moreover, the mean velocity of GPT-4 is higher than humans, suggesting that GPT-4, by default uses language representations more than humans do. The results from complexity measures show that transmitting through language has a larger effect on humans than GPT-4. Additionally, the authors use sentence embeddings to predict complexity measures in humans and GPT-4.

**Reason For Not Giving Higher Score:**

Here are some minor points.
- It would be helpful to know what prompts the authors used in the human experiment and the GPT-4 experiment. The authors mention that these were the same but do not mention the specific prompts used.
- In the Unimodal condition, the authors get language descriptions using a different set of participants but in the multi-modal condition, the authors use the language descriptions generated by participants in the task. This makes me wonder if the results in Figure 5b would still be consistent if language descriptions for the multi-modal condition were also obtained using a different set of participants. This would be a more apples-to-apples comparison of decoding complexity measures from language descriptions across both conditions.
- It is unclear why sentence embeddings were used from MPNet. MPNet seems to rank low on the MTEB leaderboard which is a benchmark to assess sentence-level representations in models. Are the results of Fig. 5B dependent on the choice of embeddings?

**Reason For Not Giving Lower Score:**

The paper is well-motivated, provides a theoretical account, and is experimentally rigorous.

**Reviewer Domain:**

cognitive science

---

### Decision · Program_Chairs · 2024-03-02

Accept (Poster)